# Robust charge-density-wave correlations in the electron-doped single-band Hubbard model

Peizhi Mai [1,2], Nathan S. Nichols [3], Seher Karakuzu[1,4], Feng Bao[5], Adrian Del Maestro [6,7,8], Thomas A. Maier [1] & Steven Johnston [6,7] ✉

There is growing evidence that the hole-doped single-band Hubbard and $t - J$ models do not have a superconducting ground state reflective of the high-temperature cuprate superconductors but instead have striped spin- and charge-ordered ground states. Nevertheless, it is proposed that these models may still provide an effective low-energy model for electron-doped materials. Here we study the finite temperature spin and charge correlations in the electron-doped Hubbard model using quantum Monte Carlo dynamical cluster approximation calculations and contrast their behavior with those found on the hole-doped side of the phase diagram. We find evidence for a charge modulation with both checkerboard and unidirectional components decoupled from any spin-density modulations. These correlations are inconsistent with a weak-coupling description based on Fermi surface nesting, and their doping dependence agrees qualitatively with resonant inelastic x-ray scattering measurements. Our results provide evidence that the single-band Hubbard model describes the electron-doped cuprates.

A key question in quantum materials research is whether or not the single-band Hubbard model describes the properties of the high-temperature (high-$T_c$) superconducting cuprates[1–3]. On the one hand, several studies have demonstrated a direct mapping between multi-orbital CuO models and effective single-band descriptions[4–7]. At the same time, quantum cluster methods have found evidence for a $d$-wave superconducting state[6,8] in the Hubbard model, with a non-monotonic $T_c$ as a function of doping that resembles the dome found in real materials. On the other hand, a growing number of state-of-the-art numerical studies on extended Hubbard and $t$-$J$ clusters have found evidence for stripe-ordered ground states for model parameters relevant to the cuprates[9–13]. While density matrix renormalization group (DMRG) simulations of multi-leg hole ($h$)-doped Hubbard ladders do

obtain a superconducting ground state for nonzero values of the next-nearest-neighbor hopping $t'$[14], its order parameter does not have the correct $d_{x^2−y^2}$ symmetry[15] found in the cuprates[16]. Conversely, DMRG calculations for six- and eight-leg $t$-$J$ cylinders obtain the correct order parameter but only on the electron ($e$)-doped side of the phase diagram[12]. These results cast significant doubt on the long-held belief that the Hubbard model describes the $h$-doped cuprates. Nevertheless, hope remains that it may capture the $e$-doped materials.

From an experimental perspective, charge-density-wave (CDW) correlations have been established as a ubiquitous feature of the underdoped cuprates[17,18]. Initially observed by inelastic neutron scattering in the form of intertwined spin and charge stripes[19], short-range CDW correlations have now been reported in nearly all families of

[1]Computational Sciences and Engineering Division, Oak Ridge National Laboratory, Oak Ridge, TN 37831-6494, USA. [2]Department of Physics and Institute of Condensed Matter Theory, University of Illinois at Urbana-Champaign, Urbana, IL 61801, USA. [3]Data Science and Learning Division, Argonne National Laboratory, Argonne, IL 60439, USA. [4]Center for Computational Quantum Physics, Flatiron Institute, 162 5th Avenue, New York, NY 10010, USA. [5]Department of Mathematics, Florida State University, Tallahassee, FL 32306, USA. [6]Department of Physics and Astronomy, The University of Tennessee, Knoxville, TN 37996, USA. [7]Institute of Advanced Materials and Manufacturing, The University of Tennessee, Knoxville, TN 37996, USA. [8]Min H. Kao Department of Electrical Engineering and Computer Science, University of Tennessee, Knoxville, TN 37996, USA. ✉e-mail: sjohn145@utk.edu

cuprates using scanning tunneling microscopy[20,21] and resonant inelastic x-ray scattering (RIXS)[22–33]. Importantly, these CDW correlations persist up to high temperatures, particularly on the $e$-doped side of the phase diagram[25,28–30].

Given their ubiquity, these CDW correlations must be accounted for by any proposed effective model for the cuprates. Evidence for charge modulations, both in the form of unidirectional stripe correlations or short-range CDW correlations, has now been found in a variety of finite-temperature quantum Monte-Carlo (QMC) simulations of the $h$-doped Hubbard model[13,34–38]. These simulations are generally restricted to high temperatures by the Fermion sign problem[10,34,35] (except for very recent constrained path QMC calculations[38]) and focus on the $h$-doped model. However, the observed cuprate CDWs exhibit a significant electron-hole asymmetry. On the $h$-doped side, they can intertwine with spin-density modulations to form stripes while they coexist with uniform antiferromagnetic (AFM) correlations on the $e$-doped side[25,28,30]. These differences have raised questions on whether the $e$- and $h$-doped CDWs share a common origin[29,30]. Addressing these questions requires detailed calculations for the $e$-doped Hubbard model, which analyze the spin and charge correlations. To date, such calculations have not been performed.

Here, we provide insights into this question by studying the spin and charge correlations of the $e$-doped two-dimensional Hubbard model and contrasting them with the $h$-doped case, using the dynamical cluster approximation (DCA)[39] and a nonperturbative QMC cluster solver[40] (see Methods). Working on large ($16 \times 4$) rectangular clusters that are wide enough to support large-period stripe correlations if they are present[34], we vary the electron density $\langle n \rangle$ across both sides of the phase diagram to contrast the correlations in each case. Our calculations uncover robust two-component CDW correlations on the $e$-doped side, which consists of superimposed checkerboard (0.5, 0.5) (in the unit of $2\pi/a$ implied for all vectors in $k$ space) and unidirectional $\mathbf{Q}_{\mathrm{CDW}} = (\pm \delta_c, 0)$ components. These CDW correlations appear to be decoupled from any stripe-like modulations of the spins and instead coexist with short-range antiferromagnetic correlations. This behavior is in direct contrast to the $h$-doped case, where we find evidence for fluctuating stripe correlations in both the charge and spin degrees of freedom[34]. Our results agree with experimental observations on the $e$-doped cuprates, including the observed doping dependence of $\mathbf{Q}_{\mathrm{CDW}}$. This supports the notion that the single-band Hubbard model captures the $e$-doped side of the high-$T_c$ phase diagram.

## Results

Figure 1 compares the static ($\omega = 0$) charge $\chi_c(\mathbf{Q}, 0)$ and spin $\chi_s(\mathbf{Q}, 0)$ susceptibilities for the $h$- ($\langle n \rangle = 0.8$) and $e$-doped ($\langle n \rangle = 1.2$) Hubbard model with $U/t = 6$, $t'/t = -0.2$, and varying temperature (see Supplementary Note 1 for error information of the lowest temperature results). In the $h$-doped case (Fig. 1a, c, e), unidirectional charge and spin stripes form as the temperature is lowered, consistent with prior finite-temperature studies[34,35,38]. These correlations manifest as incommensurate peaks in the static susceptibility at wave vectors $\mathbf{Q}_c = (\pm \delta_c, 0)$ and $\mathbf{Q}_s = (0.5 \pm \delta_s, 0.5)$ for the charge and spin channels, respectively. For the spin channel in Fig. 1e, the dashed line shows a fit of the lowest temperature data using two Lorentzian functions. Figure 1c plots the charge correlations for the $h$-doped case along the ($Q_x$, 0.5) direction, where we observe a weak double-peak structure emerging at the lowest accessible temperature. This modulation is weaker than the $\mathbf{Q}_c$ structure in Fig. 1a, such that the charge correlations are predominantly stripe-like.

We observe qualitatively different correlations in the $e$-doped case shown in Fig. 1b, d, f. At high temperature ($\beta \lesssim 6/t$), $\chi_c(\mathbf{Q}, 0)$ has a single broad peak centered at $\mathbf{q} = (0, 0)$, which can again be decomposed into two incommensurate Lorentzian peaks centered at $\pm \delta_c$, indicative of a unidirectional charge stripe. As the temperature is lowered, these peaks sharpen and become discernible without fits while the $\mathbf{q}$-independent background remains constant. The charge correlations also have a relatively temperature-independent (0.5, 0.5) component (Fig. 1e) of similar strength as the stripe-like charge correlations. In contrast to the $h$-doped case, we find no indication of a spin-stripe at this doping; the spin susceptibility has a single peak centered at (0.5, 0.5) for all accessible temperatures.

Comparing panels a and b, we see that the charge stripe correlations in the $h$- and $e$-doped systems develop differently as the temperature decreases. In the $h$-doped case (Fig. 1a), the incommensurate peaks grow while $\delta_c$ shifts to smaller values as the temperature decreases[34]. In the $e$-doped case, the incommensurate peaks grow (Fig. 1b) while the value of $\chi_c(\mathbf{Q}, 0)$ near zone center is suppressed, resulting in well-defined peaks centered at $\approx(\pm 0.25, 0)$. In addition, the (0.5, 0.5) component is significantly stronger in the $e$-doped case (Fig. 1d). However, since the height of the incommensurate peaks in Fig. 1b does not appear to level off, the stripe correlations could dominate at lower temperatures.

The corresponding correlation functions in real-space, obtained at the lowest accessible temperatures ($\beta t = 6$ and 10 for the $h$- and $e$-doped cases, respectively) are plotted in Fig. 2. (The data for the $h$-doped case are regenerated from Fig. 1D of ref. 34 with error bars.) The charge correlations in the $h$-doped case (Fig. 2a) only show a stripe pattern. In contrast, the $e$-doped case (Fig. 2b) has a clear short-range (0.5, 0.5) checkerboard-like pattern near $\mathbf{r} = 0$, superimposed over a stripe-like component. A similar (but weaker) pattern is also observed at a higher temperature in the determinantal quantum Monte-Carlo simulation for the $e$-doped Hubbard model (see Supplementary Note 2).

Figure 2c, d show the staggered spin correlation function for the $h$- and $e$-doped cases, respectively. Here, the blue region in the middle represents one AFM domain, while the red region on both sides indicates neighboring AFM domains with the opposite phase. (Note that the staggered spin correlation function that is plotted contains an additional negative sign on the B-sublattice as explained in the Methods section.) Consistent with Fig. 1 and as observed before[34], the $h$-doped case has a clear stripe pattern whose period is roughly twice that of the charge modulations. In contrast, the $e$-doped model is dominated by short-range AFM correlations with only a single phase inversion appearing at longer distances. To summarize, the $h$-doped system has intertwined spin and charge stripe correlations, while the $e$-doped system manifests CDW correlation with stripe-like $\mathbf{Q} = (\delta_c, 0)$ and checkerboard (0.5, 0.5) components and nearly uniform AFM spin correlations.

Figure 3 examines how the stripe component of the charge correlations develops for the $e$-doped case with density and $t'$ for a fixed inverse temperature $\beta = 8/t$. Figure 3a shows $\chi_c(\mathbf{Q}, 0)$ along the $(Q_x, 0)$ direction for various densities, while holding $t'/t = -0.2$ fixed. At $\langle n \rangle = 1.125$, the charge susceptibility has a single peak centered at $\mathbf{Q} = 0$. With further electron doping, the peak splits into two well-defined peaks that become discernible without fitting for $\langle n \rangle \geq 1.2$. At the same time, the uniform background grows with doping due to the increased metallicity in the system[35]. Figure 3b presents the same quantity for various values of $t'$, while fixing $\langle n \rangle = 1.2$. At $t'/t = -0.1$, the charge susceptibility again has a single broad peak at $\mathbf{Q} = 0$. As $|t'|$ increases, the middle peak is suppressed, leading to the appearance of a pair of incommensurate peaks. At the same time, the uniform background remains almost unchanged.

To assess these trends quantitatively, we fit the curves in Fig. 3a with three Lorentzian functions centered at $Q_x = 0$ and $\pm Q_c$ plus a constant background to extract the wave vector $\delta_c = Q_c$. As shown in Fig. 3c, $\delta_c$ increases approximately linearly as a function of the doping $\rho = \langle n \rangle - 1$. This trend agrees with experimental observations for the $e$-doped cuprates[28]; however, the observed $\mathbf{Q}_{\mathrm{cdw}}(\rho)$ curve is shifted to lower doping levels relative to our results. We also extracted $\delta_c$ as a

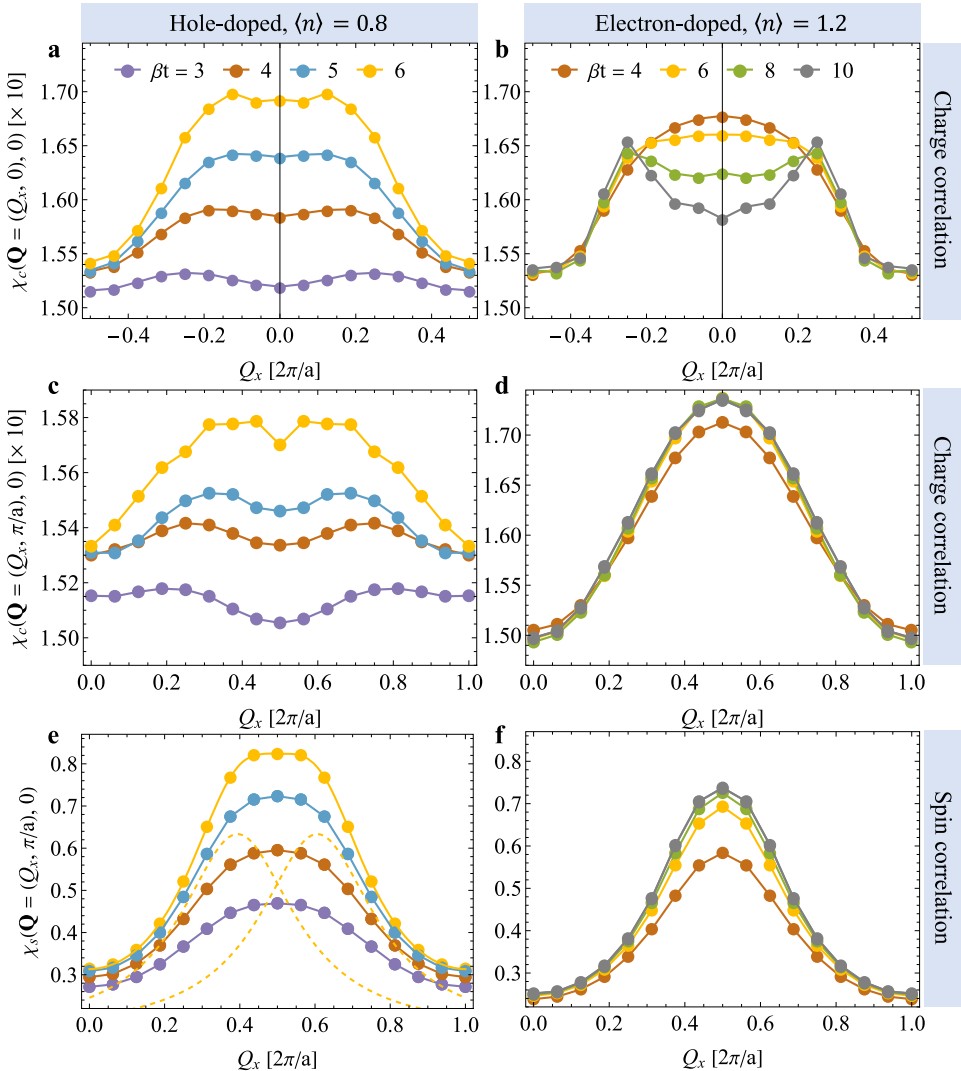

**Fig. 1 | Static spin and charge correlations in the doped single-band Hubbard model. a**, **c** show the static charge susceptibility $\chi_c(\mathbf{Q}, 0)$ along the $(Q_x, 0)$ and $(Q_x, 0.5)$ (in the unit of $2\pi/a$) directions, respectively, for the $h$-doped system with $t'/t = -0.2$ and $\langle n \rangle = 0.8$. **e** shows the corresponding static spin susceptibility $\chi_s(\mathbf{Q}, 0)$ along the $\mathbf{Q} = (Q_x, 0.5)$ direction. The yellow dashed lines show incommensurate peaks obtained from fitting multiple Lorentzian functions plus a constant background to the $\beta t = 6$ data. (The constant contribution is not shown.) **b**, **d**, **f** show the corresponding results for the $e$-doped case with $t'/t = -0.2$ and $\langle n \rangle = 1.2$. These results were obtained using a $16 \times 4$ cluster, and the inverse temperatures $\beta = 1/T$ are reported in units of $1/t$.

function of $t'$, as shown in Fig. 3d, where we find that $\delta_c$ linearly shifts to larger values with $|t'|$. Extrapolating the experimental data in Fig. 3c to $\rho = 0.2$ yields $Q_{\text{cdw}} \approx 0.32$ r.l.u., which corresponds to $|t'/t| \approx 0.4$ in our model. This value is much larger than the typical values used to model the $e$-doped cuprates using the single-band model.

RIXS experiments have found that the CDW in the $e$-doped cuprates has a significant dynamical component[25,28,30]. To compare with these measurements, we show in Fig. 4 the dynamical spin and charge structure functions $S(\mathbf{Q}, \omega)$ and $N(\mathbf{Q}, \omega)$ obtained using a parameter-free differential evolution analytic continuation (DEAC) algorithm[41]. (A comparison of the DEAC results to those obtained with more conventional techniques is provided in Supplementary Note 3.) The dynamic spin structure factor $S(\mathbf{Q}, \omega)$ displays the typical persistent antiferromagnetic paramagnon spectrum obtained with other QMC methods[42], with large spectral weight at $\mathbf{Q} = (0.5, 0)$ at an energy near $\omega \approx t$ and a downward dispersion towards $\mathbf{Q} = 0$. The dynamic charge structure factor exhibits similar behavior with a large spectral weight near the zone edge and a downward dispersion towards the zone center. Still, the spectral weight is concentrated at higher energies.

As an approximation of the predicted RIXS intensity, Fig. 4c plots a superposition of $S(\mathbf{Q}, \omega)$ and $N(\mathbf{Q}, \omega)$. Due to the vastly different energy scales in the spin and charge correlations, one sees two distinct upward dispersing branches, i.e., a low-energy branch due to the spin correlations and a high-energy branch due to the charge correlations. Since the spin correlations have a much larger amplitude, the low-energy branch has a stronger overall intensity, while the higher energy (charge) branch is muted. The behavior for the dynamical correlations near zone center agrees well with the data reported in ref. 30. Our results for the higher energy charge excitations call for future RIXS measurements in this regime; however, the high-energy portion of the charge excitations may overlap with the intra-atomic orbital excitations on the Cu atom (the so-called $dd$-excitations), which are commonly observed at the Cu $L$-edge[43].

## Discussion
The correlations in the $e$-doped case cannot be attributed solely to the weak-coupling Lindhard physics[44]. For example, as discussed in Supplementary Note 4, if we adjust the effective $U$ in a random-phase approximation (RPA) to match the charge correlations $\chi_c(Q_c, 0)$ then

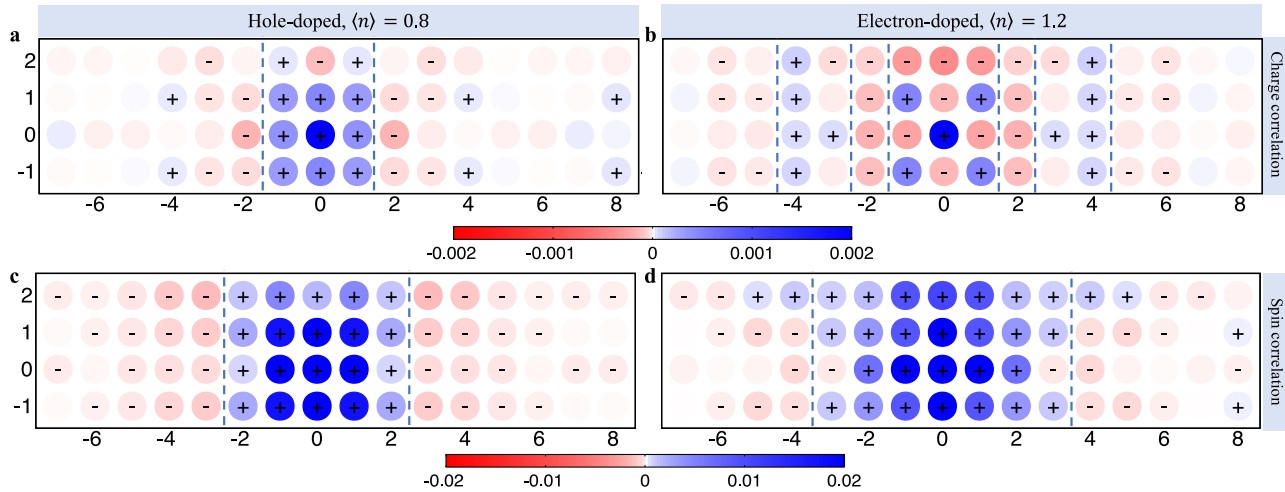

**Fig. 2 | Static spin and charge correlations in real-space. a** $\chi_c(\mathbf{r}, 0)$ for the $e$-doped system ($t' = -0.2t, \langle n \rangle = 1.2$) at the lowest accessible inverse temperature $\beta t = 10$. **b** $\chi_c(\mathbf{r}, 0)$ for the $h$-doped system ($t' = -0.2t, \langle n \rangle = 0.8$) at the lowest accessible inverse temperature $\beta t = 6$. **c**, **d** show the staggered spin-spin correlations $\chi_{s,stag}(\mathbf{r}, 0)$ for the $h$- and $e$-doped cases, respectively. $+$ and $-$ signs indicate the sign of the correlations whose absolute mean is larger than two standard errors. The dashed lines indicate the approximate nodes in the spin and charge stripe modulations.

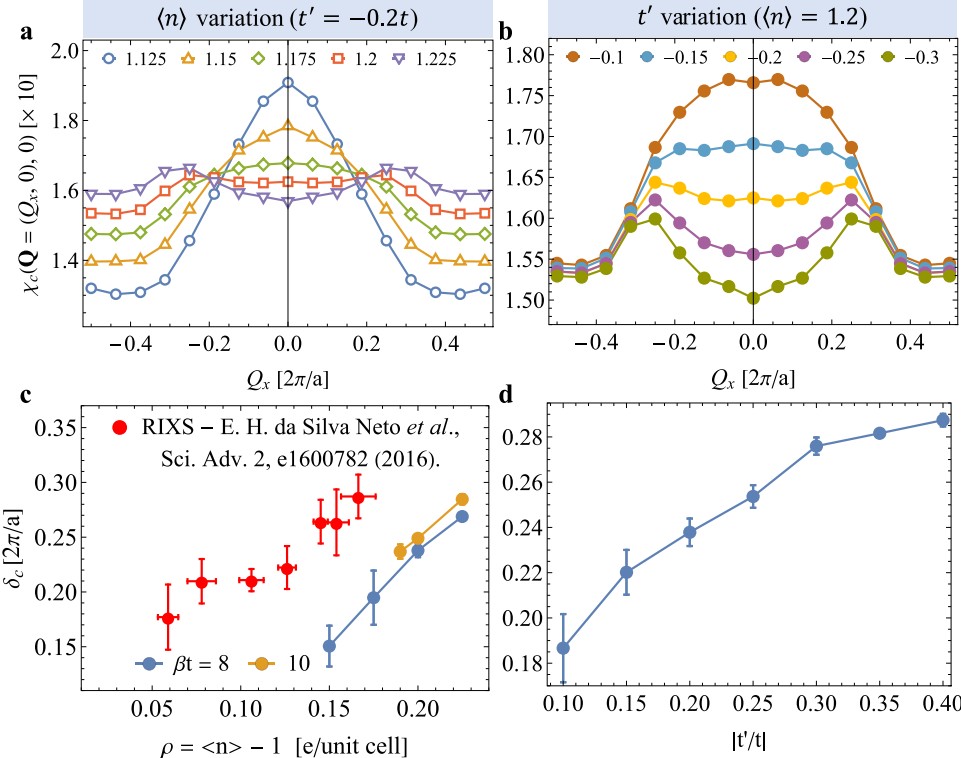

**Fig. 3 | Evolution of the charge-density-wave correlations with model parameters. a**, **b** show $\chi_c(\mathbf{Q}, 0)$ for the $e$-doped case along the $(Q_x, 0)$ direction at $\beta = 8/t$. **a** shows the effect of different electron fillings $\langle n \rangle$ at $t' = -0.2t$ while **b** shows the effect of varying $t'$ for fixed $\langle n \rangle = 1.2$. **c** The evolution of the incommensurate CDW peak $\mathbf{Q}_{cdw} = (\delta_c, 0)$ with doping, obtained from fitting the spectra in **a** with three Lorentzian functions and a constant background. For comparison, we also plot the measured values of $\mathbf{Q}_{cdw}$ extracted from RIXS experiments[28]. **d** The evolution of the incommensurate CDW peak as a function of $t'$, obtained from fitting the data shown in **b**. All results were obtained on a $16 \times 4$ cluster with $U = 6t$ and $\beta t = 8$ except for the $\beta t = 10$ results in **c**. The error bars in **c**, **d** are standard deviations of errors from fitting the data to three Lorentzian functions plus a constant background.

the predicted correlations at (0.5, 0.5) are much stronger than those reported here. Similarly, we do not resolve any peak structure in $\chi_s(\mathbf{Q}, 0)$ along the $(Q_x, 0)$ direction as one would expect in such a weak-coupling framework. While the peak positions predicted by RPA are close to the values reported here for $\langle n \rangle = 1.2$, the resulting temperature, doping, and $t'$ evolution is inconsistent with the observed behavior (see Supplementary Note 4).

We have demonstrated that the CDW correlations observed in DCA simulations of the single-band Hubbard model have a pronounced particle-hole asymmetry. Our results for the $e$-doped system are also in qualitative agreement with RIXS experiments on Nd$_{2-x}$Ce$_x$CuO$_4$[25,30]; however, there are some notable quantitative differences. For example, our predicted $\delta_c(\rho)$ dependence is shifted to higher electron doping in comparison to experiment. This

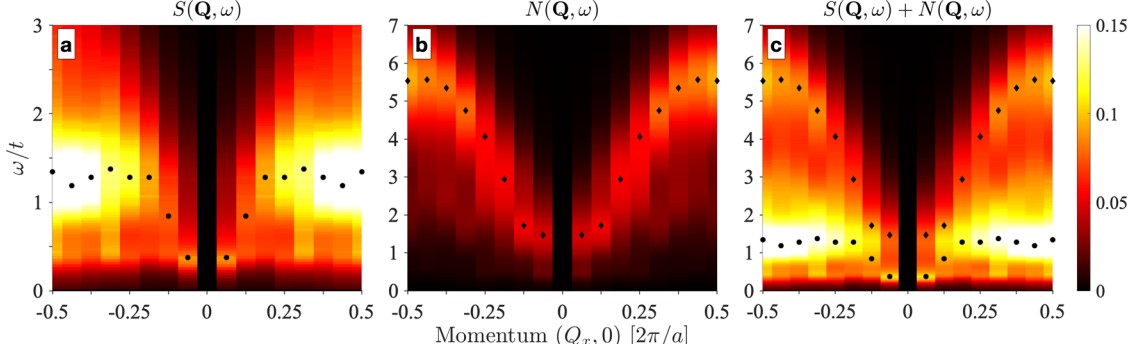

**Fig. 4 | Dynamical spin and charge structure factors for the *e*-doped model.**
**a**, **b** show the dynamical spin $S(\mathbf{Q}, \omega)$ and charge $N(\mathbf{Q}, \omega)$ structure factors, respectively, along the $\mathbf{Q} = (Q_x, 0)$ direction. Results are shown for an *e*-doped model with $t'/t = -0.2$ and $\langle n \rangle = 1.2$, obtained on a $16 \times 4$ cluster with $U = 6t$ and $\beta/t = 10$. **c** shows the sum of the two as a crude approximation for Cu *L*-edge RIXS spectra. The black circles and diamonds indicate the locations of the maxima in $S(\mathbf{Q}, \omega)$ and $N(\mathbf{Q}, \omega)$, respectively. All panels are plotted on the same color scale, as indicated on the right.

discrepancy may be related to challenges in determining the carrier concentration in the $CuO_2$ planes. ARPES measurements of $Pr_{1.3-x}La_{0.7}Ce_xCuO_{4-\delta}$ (PLCCO), for example, have suggested that the doped electron concentration of $CuO_2$ plane can be larger than the Ce concentration $x$ by up to $0.08$ $e$/Cu, depending on the annealing method[45]. This discrepancy is comparable to the shift observed in Fig. 3c. Adjusting the value of $t'$ could also partially account for this difference; single-band fits to the measured Fermi surface of PLCCO estimate $|t'/t| \approx 0.34 - 0.4$. Finally, the periodicity of the charge modulations may be affected by the DCA mean-field[34]. In the *h*-doped case, DCA and DQMC predict different stripe periods for the same model parameters. Nevertheless, our calculations reproduce the qualitative doping dependence observed in the real materials and support the notion that the single-band Hubbard model captures the physics of the *e*-doped cuprates.

## Methods
### The model
We consider the single-band Hubbard model on a two-dimensional square lattice. The Hamiltonian is

$$H = -\sum_{\mathbf{i,j},\sigma} t_{\mathbf{i,j}} c^{\dagger}_{\mathbf{i},\sigma} c_{\mathbf{j},\sigma} - \mu \sum_{\mathbf{i},\sigma} n_{\mathbf{i},\sigma} + U \sum_{\mathbf{i}} n_{\mathbf{i},\uparrow} n_{\mathbf{j},\downarrow}. \quad (1)$$

Here, $c^{\dagger}_{\mathbf{i},\sigma}$ ($c_{\mathbf{i},\sigma}$) creates (annihilates) a spin-$\sigma$ ($= \uparrow, \downarrow$) electron at site $\mathbf{i} = a(i_x, i_y)$, where $a = 1$ is the lattice constant, $t_{\mathbf{i,j}}$ is the hopping integral between sites $\mathbf{i}$ and $\mathbf{j}$, $\mu$ is the chemical potential, and $U$ is the Hubbard repulsion. To model the cuprates, we set $t_{\mathbf{i,j}} = t$ and $t'$ for nearest and next-nearest-neighbor hopping, respectively, and zero otherwise, and take $U/t = 6$ unless stated otherwise.

### The dynamical cluster approximation
We study the model in Eq. (1) using DCA[8,39,46] as implemented in the DCA++ code[47]. The DCA represents the infinite bulk lattice in the thermodynamic limit by a finite-size cluster embedded in a self-consistent dynamical mean-field. The intra-cluster correlations are treated exactly, while the mean-field approximates the inter-cluster degrees of freedom. We use rectangular $N_c = 16 \times 4$ clusters that are large enough to support spin and charge stripe correlations if they are present in the model[34].

Assuming short-ranged correlations, the self-energy $\Sigma(\mathbf{k}, i\omega_n)$ can be approximated by the cluster self-energy $\Sigma(\mathbf{K}, i\omega_n)$, where $\mathbf{K}$ are the cluster momenta. We obtain the coarse-grained single-particle Green

function

$$\bar{G}(\mathbf{K}, i\omega_n) = \frac{N_c}{N} \sum_{\mathbf{k}'} G(\mathbf{K} + \mathbf{k}', i\omega_n)$$
$$= \frac{N_c}{N} \sum_{\mathbf{k}'} \frac{1}{i\omega_n + \mu - \varepsilon(\mathbf{K} + \mathbf{k}') - \Sigma(\mathbf{K}, i\omega_n)}, \quad (2)$$

by averaging the lattice Green function $G(\mathbf{k}, i\omega_n)$ over the $N/N_c$ momenta $\mathbf{k}'$ in a patch about the cluster momentum $\mathbf{K}$. ($N$ and $N_c$ are the number of site in the lattice and cluster, respectively.) In this way, the bulk problem is reduced to a finite-size-cluster problem.

We solve the cluster problem with the continuous-time, auxiliary-field quantum Monte-Carlo algorithm (CT-AUX)[40,48]. The expansion order of the CT-AUX QMC algorithm is typically 500–1500, depending on the temperature and value of $t'$. Depending on the value of the average fermion sign for a given parameter set, we measure $1-5 \times 10^8$ samples for the correlation functions. Six to eight iterations of the DCA loop are typically needed to obtain good convergence for the DCA cluster self-energy.

To study the spin and charge correlations, we measure the static ($\omega = 0$) real-space spin-spin correlation function

$$\chi_s(\mathbf{r}, 0) = \frac{1}{N} \sum_{\mathbf{i}} \int_0^{\beta} \left\langle \hat{S}^z_{\mathbf{r+i}}(\tau) \hat{S}^z_{\mathbf{i}}(0) \right\rangle d\tau \quad (3)$$

and density-density correlation function

$$\chi_c(\mathbf{r}, 0) = \frac{1}{N} \sum_{\mathbf{i}} \int_0^{\beta} \left[ \langle n_{\mathbf{r+i}}(\tau) n_{\mathbf{i}}(0) \rangle - \langle n_{\mathbf{r+i}}(\tau) \rangle \langle n_{\mathbf{i}}(0) \rangle \right] d\tau. \quad (4)$$

Here, $\mathbf{r} = (r_x, r_y)a$ and $\mathbf{i} = (i_x, i_y)a$ are lattice vectors and $\hat{S}^z_{\mathbf{i}} = \frac{1}{2}(n_{\mathbf{i},\uparrow} - n_{\mathbf{i},\downarrow})$ and $n_{\mathbf{i}} = \sum_{\sigma} c^{\dagger}_{\mathbf{i},\sigma} c_{\mathbf{i},\sigma}$ are the local spin-$z$ and density operators of the effective cluster problem. The correlation functions $\chi_{c,s}(\mathbf{r}, 0)$ are measured directly by the cluster solver. The static momentum-space susceptibilities $\chi_c(\mathbf{Q}, 0)$ and $\chi_s(\mathbf{Q}, 0)$ are obtained by a Fourier transform. We also plot the staggered spin correlation function $\chi_{s,stag}(\mathbf{r}) = (-1)^{(r_x + r_y)} S(\mathbf{r}, 0)$ to highlight the relative phases of the antiferromagnetic domains.

### Analytic continuation
The dynamical charge $N(\mathbf{Q}, \omega)$ and spin $S(\mathbf{Q}, \omega)$ structure factors are obtained from the imaginary part of the charge and spin

susceptibilities using

$$N(\mathbf{Q},\omega) = \frac{\mathrm{Im}\chi_c(\mathbf{Q},\omega)}{1 - e^{-\beta\omega}}, \tag{5}$$

and

$$S(\mathbf{Q},\omega) = \frac{\mathrm{Im}\chi_s(\mathbf{Q},\omega)}{1 - e^{-\beta\omega}}. \tag{6}$$

The real frequency susceptibility is related to the imaginary time susceptibility by the integral equation

$$\chi_{s,c}(\mathbf{Q},\tau) = \int_0^\infty \frac{d\omega}{\pi} \frac{e^{-\tau\omega} + e^{-(\beta-\tau)\omega}}{1 - e^{-\beta\omega}} \mathrm{Im}\chi_{s,c}(\mathbf{Q},\omega). \tag{7}$$

Inverting this relationship to obtain $\mathrm{Im}\chi(\mathbf{Q},\omega)$ is an ill-posed problem. We used three independent methods to perform the analytic continuation to gauge our relative confidence in the various features. These include the method of Maximum Entropy[49], a parameter-free differential evolution algorithm[41], and stochastic optimization[50]. The results obtained using the differential evolution algorithm are shown in the main text, while the remaining results are provided in Supplementary Note 3.

## Data availability
The data in this study are available at https://doi.org/10.5281/zenodo.7829220.

## Code availability
The DCA++ code used for this project can be obtained at https://github.com/cosdis/DCA_singleband_edoped/tree/TpEqTimeAccOnGpu_onMasterb and https://doi.org/10.5281/zenodo.7830135 with DOI: 10.5281/zenodo.7830135. The DEAC code can be obtained at https://github.com/DelMaestroGroup/papers-code-DEAC. All other codes will be made available upon request.

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

## Acknowledgements
We thank M. P. M. Dean and J. Pelliciari for their valuable discussions. This work was supported by the U.S. Department of Energy, Office of Science, Office of Basic Energy Sciences, under Award Number DE-SC0022311. F.B. was supported by the Department of Energy, Office of Science, Advanced Scientific Computing Research (ASCR) progam, under Award Number DE-SC0022297. N.S.N. was supported by the Argonne Leadership Computing Facility, which is a U.S. Department of Energy Office of Science User Facility operated under contract DE-AC02-06CH11357. This research used resources from the Oak Ridge Leadership Computing Facility, which is a DOE Office of Science User Facility supported under Contract No. DE-AC05-00OR22725.

## Author contributions
P.M. performed the DCA calculations, analyzed the data, and carried out the MaxEnt analytic continuation calculations; N.S.N. and A.D.M. developed the differential evolution analytic continuation code and performed analyses with this method; F.B. developed the stochastic optimization code and performed analyses with this method. S.K. and T.A.M. developed the DCA code. T.A.M. and S.J. supervised the project; P.M., T.A.M., and S.J. wrote the paper with input from all authors.

## Competing interests
The authors declare no competing interests.
