## [Peer Review File · Nature Communications]

REVIEWER COMMENTS

Reviewer #1 (Remarks to the Author):

It has long been proposed that the single-band Hubbard model can be considered as a microscopic model to understand the physics of unconventional superconductors such as cuprates. Although growing evidence suggests that this model may not be superconducting when doped with holes, it may still be able to capture some key physics of the cuprates such as pseudo-gap phase, strange metal phase, charge and spin density wave orders. In this work, the authors focus on the finite temperature spin and charge correlations in the electron doped single-band Hubbard model on a 16×4 cluster on the square lattice using quantum Monte Carlo (QMC) dynamical cluster approximation (DCA) calculations and compare its behavior with that of the hole doped side of the phase diagram. The authors have observed a clear particle-hole asymmetry between the hole and electron doped sides as the symmetry is broken by a finite next-nearest-neighbor electron hopping t' . The authors have provided evidence for the presence of coexisting charge modulations with both checkerboard and unidirectional components that are uncorrelated with spin-density modulations in the electron doped side. Comparisons with RIXS measurements have also been made. The observation of particle-hole asymmetry is clear and consistent with previous numerical studies. The possible stronger charge ordering tendency in the electron doped side is interesting, and the paper is also well-written. However, there are some important issues which are needed to be addressed and checked before I can recommend its publication. The details are given below.

The authors have considered a negative t' compared with the nearest-neighbor electron hopping t , and studied the hole and electron doped cases when the electron density is below and above the half-filling, respectively. The latter is equivalent to the positive t' case but with electron density below half-filling considered in the previous studies after a particle-hole transformation, for which direct comparisons can be made. In the current study, the authors have shown that the pattern of charge density wave order in the electron doped side at low temperature, e.g., $\beta t = 8$ and 10 , is likely consistent with a coexisting checkerboard and unidirectional charge stripe. However, the numerical results that the charge stripe is close to insulating filled charge stripe are inconsistent with previous numerical studies of the single-band Hubbard and related t - J models on the square lattice, where the density-matrix renormalization group (DMRG) calculations suggest that the charge density wave order is unidirectional and half-filled on four-leg ladders. This is surprising because this difference is likely qualitative and can lead to a dramatic different state. For example, a state with filled charge stripe will be insulating while a state with half-filled charge stripe can be metallic or even superconducting. Moreover, there was no reported checkerboard charge density wave order in previous studies. Therefore, it will be important if the authors can clarify and explain why the DCA calculations produce likely qualitatively different results, and whether these results are reliable compared with previous studies such as QMC and DMRG.

The authors have also compared their results with RIXS experiments on related materials and observed some inconsistencies between them. The RIXS experiments in Fig.3c are more consistent with half-filled charge stripe, while the DCA results in the current study are more consistent with filled charge stripe. The authors attribute this difference to possibly different model parameters as well as limitation of DCA mean-field. However, as mentioned previously, other numerical studies have produced results that are more consistent with the RIXS experiments using the similar models and parameters. This problem needs to be addressed seriously.

Reviewer #2 (Remarks to the Author):

The authors study the Hubbard model with the DCA method with a CTQMC (CT-AUX) impurity solver. The cluster solved there are quite big, as it is necessary to study stripe order ground state or charge- and spin-susceptibility. The authors find that if the calculation is done properly, the resulting susceptibilities differ significantly from the low interaction approximations of the susceptibility. This indicates that the true ground state of the Hubbard model has charge and spin modulations, as measured with resonant inelastic x-ray scattering. Additionally, the authors claim that this modulation is not stripe-like on the e-doped side, as opposed to the h-doped results previously obtained.

The general presentation is good. I did not rerun any calculation in order to confirm, I trust the authors on the calculation part. I have few reserves on the presentation and the interpretation and I will list them below. The results are interesting, but I fail to see their significance in the context. In summary, this work is a natural extension to Ref. 10 and 34. These references only focussed on the h-doped Hubbard model whereas this work extend the simulation to the e-doped. As a byproduct, it provides a good benchmark and comparison of some results of DQMC and DCA+CTMQC. Finally, this is another good proof that any low interaction approximation of susceptibility fails to capture the physics measured by the DCA method. These are all good results, but the authors fail to highlight their significance, in my opinion.

#####

Presentation:

1. It is not clear why the ordering of the columns "hole-doped" and "electron-doped" has been inverted in Fig. 2 compared to Fig. 1. This results in an unnecessary confusing presentation. I suggest keeping the same order for both figures.

2. The notation $(Q_x, 0)$, $(Q_x, 1)$ on the figures and the contrasting notation $(Q_x, \pi/a)$ is generally confusing (even in the supplementary information, the notation (Q_x, π) is used). I suggest uniformizing this.

It would be preferable to change the axis ticks to $(0, \pi/a, 2\pi/a)$ and forget the ± 0.5 ticks in Fig. 1b for example. That way, the label $(Q_x, \pi/a)$ could be used for the x-axis label. In fact, this connects with another presentation problem. The x-axis is only one scalar value, yet, a vector in which there is a Q_x is written. We can deduce that what is meant is that the x-axis is really only Q_x , but this is not rigorous nor clear. A simpler solution, and far clearer and more rigorous would be to change the y-axis to $\chi_s(\vec{Q}=(Q_x, \pi/a), \omega = 0)$ and use $Q_x [\pi/a]$ on the x-axis.

3. Sometimes, in the same sentence, "electron-doped" is used with "h-doped". It would be better to try to stick to the same format in the same sentence/paragraph, and ideally for the whole text.

4. What is the meaning of dd-excitation?

5. I suggest replacing the sentence "(The dynamical spin structure factor $S(Q, \omega)$ and spin susceptibility $\chi_s(Q, \omega)$ satisfy a similar equation)" by the equation itself.

6. It is very interesting to have access to the codes used for this project. The link provided for both github repository give access to the very general repositories, with multiple branches. It would be necessary to have a tag for the version used for this project. I suggest a tag in the git history, but a fork of the code in separate git repositories, dedicated to the versions used in this manuscript would be even better.

7. The equation for local density operator suggests that it is possible to measure local quantity with DCA. Usually DCA is periodized, so the Green function is translationally invariant. I deduce that the local charge operator is then measured on the cluster directly. In the text, I would write, in the sentence after equation 4: "[...] is the local density operator measured on the cluster from the impurity solver."

#####

Physics:

1. There is a strong claim that the CDW seen on the e-doped case is not correlated to the SDW, as opposed to the h-doped case. I do not see this clearly from Fig.2. Indeed, Fig.2a and b both have a somewhat misleading colorbar in the sense that most dots on the far left and right have intensity below 3% of the maximum value. This means that, except from the 5 centre most columns, most of the data are noise fluctuations around zero. I do not know how much can be claimed from this and if the dotted lines can be trusted.

2. In the text, it is written:

"[...] are plotted in Fig. 2, with the data for the h-doped case reproduced from Ref. 34", which seems to suggest that Fig.2d (and potentially Fig.2c) are taken directly from Ref. 34. I have looked at Ref. 34 and no such data are found, so I deduced that my first reading was wrong and this sentence only meant that "Fig.2d is very similar to Fig.1a of Ref. 34." I would be much clearer about it. I would even add something like "This similarity act as a benchmark that bridges the results from DQMC method (used in Ref. 10 and 34) with our DCA method.". Although, there is a factor of 5 in the colorscale I cannot explain, this should also be addressed in the text.

3. The equations 3 and 4 are really the spatial average of the local cluster spin and charge susceptibility. It is not clear how accurate this averaging is. Indeed, in DCA the self-consistency is done on a periodic and translational invariant Green function. This does not imply that the original local cluster quantities are a good representation of the real physics. This needs to be addressed in the text. Perhaps the best answer to this point is that the results from Ref. 34 are well reproduced, hence this constitute a good benchmarked. Any additional comments or reference would improve the strength of the conclusion.

4. In the discussion, the authors spend some time arguing why their result differs quantitatively from the experiments. I find this portion superfluous because it is generally known and accepted that CDMFT and DCA provide qualitatively accurate results, but rarely quantitatively accurate. Indeed, the quantitative inaccuracy of the CDW peak compared to the experiment could be caused by the inaccuracy of the experimental doping value because of the annealing method. But many more causes are possible, principally the general numerical inaccuracy of the Hubbard model solved with DMFT extensions and their approximations.

RESPONSE TO REVIEWER #1

The Reviewer wrote: It has long been proposed that the single-band Hubbard model can be considered as a microscopic model to understand the physics of unconventional superconductors such as cuprates. Although growing evidence suggests that this model may not be superconducting when doped with holes, it may still be able to capture some key physics of the cuprates such as pseudo-gap phase, strange metal phase, charge and spin density wave orders. In this work, the authors focus on the finite temperature spin and charge correlations in the electron doped single-band Hubbard model on a 16×4 cluster on the square lattice using quantum Monte Carlo (QMC) dynamical cluster approximation (DCA) calculations and compare its behavior with that of the hole doped side of the phase diagram. The authors have observed a clear particle-hole asymmetry between the hole and electron doped sides as the symmetry is broken by a finite next-nearest-neighbor electron hopping t' . The authors have provided evidence for the presence of coexisting charge modulations with both checkerboard and unidirectional components that are uncorrelated with spin-density modulations in the electron doped side. Comparisons with RIXS measurements have also been made. The observation of particle-hole asymmetry is clear and consistent with previous numerical studies. The possible stronger charge ordering tendency in the electron doped side is interesting, and the paper is also well-written. However, there are some important issues which are needed to be addressed and checked before I can recommend its publication. The details are given below.

Our response: We thank the referee for their summary, their interest in our work, and for their time. Below we will address their concerns.

The Reviewer wrote: The authors have considered a negative t' compared with the nearest-neighbor electron hopping t , and studied the hole and electron doped cases when the electron density is below and above the half-filling, respectively. The latter is equivalent to the positive t' case but with electron density below half-filling considered in the previous studies after a particle-hole transformation, for which direct comparisons can be made. In the current study, the authors have shown that the pattern of charge density wave order in the electron doped side at low temperature, e.g., $\beta t = 8$ and 10 , is likely consistent with a coexisting checkerboard and unidirectional charge stripe. However, the numerical results that the charge stripe is close to insulating filled charge stripe are inconsistent with previous numerical studies of the single-band Hubbard and related t - J models on the square lattice, where the density-matrix renormalization group (DMRG) calculations suggest that the charge density wave order is unidirectional and half-filled on four-leg ladders. This is surprising because this difference is likely qualitative and can lead to a dramatic different state. For example, a state with filled charge stripe will be insulating while a state with half-filled charge stripe can be metallic or even superconducting. Moreover, there was no reported checkerboard charge density wave order in previous studies. Therefore, it will be important if the authors can clarify and explain why the DCA calculations produce likely qualitatively different results, and whether these results are reliable compared with previous studies such as QMC and DMRG.

Our response: We respectfully disagree that our results are inconsistent with prior numerical work. We are unaware of any calculations on the electron-doped side that contradict our results. As we have outlined in our introduction, the vast majority of the studies on the single-band Hubbard model have focused on the hole-doped side of the phase diagram.

From the DMRG perspective, only one paper presents results for the single-band Hubbard model on the electron-doped side of the phase diagram.¹ However, this study does not show real-space charge correlations or analyze the charge correlations in a way that allows us to determine if they see the checkerboard correlations reported in our work. There is a recent DMRG study² on the electron-doped t - J model that does show unidirectional stripes. However, the correspondence between the Hubbard and t - J model becomes exact only in the infinite U limit, and our value of $U = 6t$ is far from that limit. For $U = 6t$, we expect that doubly occupied sites still play a significant role in shaping the charge fluctuations. In contrast, in the t - J

¹Y. F. Jiang *et al.*, *Phys. Rev. Res.* **2**, 033073 (2020).

²Jiang *et al.*, *PNAS* **118**, e2109978118 (2021)

Figure 1: **Static charge correlations in real-space from DQMC simulations compared to the DCA results.** **a** and **b** plot $\chi_c(\mathbf{r}, 0)$ for the e -doped system ($\langle n \rangle = 1.2$) at $t'/t = -0.2$ and -0.3 , respectively. The other parameters are the inverse temperature $\beta = 4.5/t$ and $U/t = 6$. + and - signs indicate the sign of the correlations whose absolute mean is larger than two standard errors. The dashed lines indicate the approximate nodes in charge stripe modulations. Panels **c** and **d** show the corresponding DCA results for $\beta = 8$.

model, these states are projected out, which could suppress the (π, π) component of the charge modulations. For this reason, one cannot assert with any certainty that the t - J model results contradict ours.

The recent diagrammatic Monte Carlo work by Šimković IV *et al.*³ and the constrained path auxiliary field QMC study by Xiao *et al.*⁴ both focus exclusively on the hole-doped Hubbard model, and the latter did not include t' .

Turning to DQMC, one study⁵ of the single-band Hubbard model examines the stripe correlations on the hole- and electron-doped sides of the phase diagram. However, that paper could only resolve the spin stripes and presented no results for the charge correlations. A subsequent study by the same group, together with some of us, reported DQMC results for the charge correlations in the single-band model⁶ but only examined the hole-doped side of the phase diagram. We also note that both DQMC studies were restricted to temperatures nearly a factor of two larger than the ones considered in our work.

To summarize, no published numerical results for the combined spin and charge correlations of the electron-doped Hubbard model contradict our results. We agree that it is important to check consistency with other numerical results. To that end, we have conducted additional DQMC simulations for the charge correlations in the electron-doped Hubbard model. Our DQMC calculations were conducted on 16×4 clusters with $U/t = 6$, $\langle n \rangle = 1.2$, and $t'/t = -0.2$ and -0.3 at a higher temperature $\beta = 4.5/t$. The spin correlations (not shown) are very similar to those of E. W. Huang *et al.*, [npj Quant. Mat. **3**, 22 (2018)] and reveal uniform antiferromagnetism. The charge correlations, shown in Fig. 1, reveal both a unidirectional and a (π, π) component to the charge modulations. Both results are in close agreement with our DCA results. These new results, now included in our revised Supplementary Note 1, demonstrate that our DCA results are consistent with other non-perturbative methods. We also note that we do not see any checkerboard modulations on the hole-doping side, which is consistent with other numerical methods at the same doping level. Therefore, we are confident that our results are compatible with the existing literature where it overlaps.

The Reviewer wrote: The authors have also compared their results with RIXS experiments on related materials and observed some inconsistencies between them. The RIXS experiments in Fig. 3c are more consistent with half-filled charge stripe, while the DCA results in the current study are more consistent with filled charge stripe. The authors attribute this difference to possibly different model parameters as well as

³F. Šimković IV, R. Rossi, and M. Ferrero, *Phys. Rev. Res.* **4**, 043201 (2022).

⁴B. Xiao *et al.*, *Phys. Rev. X* **13**, 011007 (2023).

⁵E. W. Huang *et al.*, *npj Quant. Mat.* **3**, 22 (2018)

⁶E. W. Huang *et al.*, *arXiv:2202.08845* (2022).

limitation of DCA mean-field. However, as mentioned previously, other numerical studies have produced results that are more consistent with the RIXS experiments using the similar models and parameters. This problem needs to be addressed seriously.

Our response: To our knowledge, no group has attempted to quantitatively compare the doping dependence of the charge correlations obtained from non-perturbative numerics against RIXS measurements. This is particularly true on the electron-doped side of the phase diagram because the charge correlations of the electron-doped Hubbard model have yet to be widely examined, as outlined above. Can the Reviewer please provide the reference for the more robust agreement they are citing here?

We also do not believe that one can say that the RIXS measurements on the electron-doped side are consistent with half- or fully-filled stripes. As we discuss in our paper, there is currently a debate in the literature as to whether or not the dopant concentration x can be mapped directly onto the doped carrier concentration ρ of the CuO_2 plane in the electron-doped cuprates.

RESPONSE TO REVIEWER #2

The Reviewer wrote: The authors study the Hubbard model with the DCA method with a CTQMC (CT-AUX) impurity solver. The cluster solved there are quite big, as it is necessary to study stripe order ground state or charge- and spin-susceptibility. The authors find that if the calculation is done properly, the resulting susceptibilities differ significantly from the low interaction approximations of the susceptibility. This indicates that the true ground state of the Hubbard model has charge and spin modulations, as measured with resonant inelastic x-ray scattering. Additionally, the authors claim that this modulation is not stripe-like on the e-doped side, as opposed to the h-doped results previously obtained.

Our response: We thank the Reviewer for their time, for their summary, and for their interest in our work. Before proceeding, we would like to clarify one point. Our claim is that the charge correlations are not intertwined with the spin modulations in the sense that no spin modulation is forming over the uniform antiferromagnetism. In that sense they are not stripe-like. However, our results do show that the charge modulations have a unidirectional component superimposed over the $\mathbf{Q} = (\pi/a, \pi/a)$ component. The unidirectional component is reminiscent of stripe-like correlations if one does not define stripes as the *intertwined* spin- and charge-order.

The Reviewer wrote: The general presentation is good. I did not rerun any calculation in order to confirm, I trust the authors on the calculation part. I have few reserves on the presentation and the interpretation and I will list them below. The results are interesting, but I fail to see their significance in the context. In summary, this work is a natural extension to Ref. 10 and 34. These references only focussed on the h-doped Hubbard model whereas this work extend the simulation to the e-doped. As a byproduct, it provides a good benchmark and comparison of some results of DQMC and DCA+CTMQC. Finally, this is another good proof that any low interaction approximation of susceptibility fails to capture the physic measured by the DCA method. These are all good results, but the authors fail to highlight their significance, in my opinion.

Our response: We thank the referee for this comment as it gives us an opportunity to clarify the significance of our work.

There is a growing body of literature suggesting that the two-dimensional single-band Hubbard model does not describe the low-temperature properties of the hole-doped cuprates. For example, multiple state-of-the-art numerical methods have concluded that ground state of the model is characterized by intertwined spin- and charge-strips rather than a d -wave superconducting state. Conversely, a recent DMRG study⁷ of the electron-doped t - J model has obtained a superconducting ground state with the correct $d_{x^2-y^2}$ symmetry on large multi-leg ladders. This result gives us hope that the single-band Hubbard model may still be a good effective model for the electron-doped cuprates, even if it does not describe the hole-doped cuprates. However, to demonstrate this conclusively, one must also show that the spin and charge correlations observed in the electron-doped cuprates are captured by the single-band models. This, in turn, requires systematic calculations in this region of the phase diagram with comparisons to experiments.

As we have outlined to Reviewer #1, the electron-doped side of the phase diagram has not been widely explored using most state-of-the-art numerical methods. Our work fills this crucial gap and provides systematic results for the spin and charge correlations in the electron-doped model at finite temperatures and in the thermodynamic limit. Importantly, our results qualitatively capture the trends observed by RIXS in the real materials, including their high-temperature doping dependence. This result is significant because it demonstrates that indeed the Hubbard model can provide a fairly accurate description of the finite-temperature properties of the electron-doped cuprates. This fact has not been established by any prior studies.

We also note that the current work reaches nearly two times smaller temperatures than those accessed in Refs. [10] and [34]. For example, our $\beta t = 10$ corresponds to $T \approx 290 - 350$ K if we adopt the standard value of $t = 250 - 300$ meV. Reaching these temperatures is significant because it means we can now perform calculations at temperatures overlapping with existing RIXS experiments on the electron-doped side of the phase diagram. Such comparisons have not been possible on the hole-doped side, where DQMC and DCA have only been able to reach $\beta t = 4.5$ ($T \approx 644$ K) and 6 ($T \approx 483$ K), respectively, after adopting $t = 250$ meV.

⁷Jiang *et al.*, PNAS 118, e2109978118 (2021).

In response to the Reviewer’s comments, we have revised our manuscript in several places to better articulate the significance of our results for a general reader.

The Reviewer wrote: Presentation: 1. It is not clear why the ordering of the columns “hole-doped” and “electron-doped” has been inverted in Fig. 2 compared to Fig. 1. This result in an unnecessary confusing presentation. I suggest keeping the same order for both figures.

Our response: We thank the Reviewer for this comment. We had initially ordered the panels to follow the sequence in which we discussed them. However, we agree that this could lead to some confusion. In the revised manuscript, we have taken the Reviewer’s suggestion and kept the same ordering of the columns in Fig. 1 and Fig. 2. We have then revised the text accordingly.

The Reviewer wrote: 2. The notation $(Q_x, 0)$, $(Q_x, 1)$ on the figures and the contrasting notation $(Q_x, \pi/a)$ is generally confusing (even in the supplementary information, the notation (Q_x, π) is used). I suggest uniformizing this.

Our response: We thank the reviewer for pointing this out. We have updated the manuscript and stuck to the notation $(Q_x, 0.5)$, $(Q_x, 0)$ in the unit of $2\pi/a$ through the text.

The Reviewer wrote: It would be preferable to change the axis ticks to $(0, \pi/a, 2\pi/a)$ and forget the ± 0.5 ticks in Fig.1b for example. That way, the label $(Q_x, \pi/a)$ could be used for the x-axis label. In fact, this connects with another presentation problem. The x-axis is only one scalar value, yet, a vector in which there is a Q_x is written. We can deduce that what is meant is that the x-axis is really only Q_x , but this is not rigorous nor clear. A simpler solution, and far clearer and more rigorous would be to change the y-axis to $\chi_s(\vec{Q} = (Q_x, \pi/a), \omega = 0)$ and use $Q_x[\pi/a]$ on the x-axis.

Our response: We took the suggestions to change the axis labels.

The Reviewer wrote: 3. Sometimes, in the same sentence, “electron-doped” is used with “h-doped”. It would be better to try to stick to the same format in the same sentence/paragraph, and ideally for the whole text.

Our response: We thank the Reviewer for bringing this to our attention. We have revised the text to use the same notation throughout.

The Reviewer wrote: 4. What is the meaning of dd-excitation?

Our response: The term “dd-excitation” is used in the RIXS literature to describe intra-atomic, inter-orbital excitations in 3d and 4d transition metal systems. In the cuprates, they correspond to excitations where the Cu $3d_{x^2-y^2}$ hole is excited into one of the other four Cu 3d orbitals on the same atom during the scattering event. These excitations typically appear around 1 – 1.5 eV, which overlaps with the upper portion of the charge excitations predicted by our calculations assuming the typical value $t \approx 250 - 300$ meV. In response to the reviewer’s question, we have modified the text

Our results on the higher-energy charge excitations call for future RIXS measurements in this regime; however, the high-energy portion of the charge excitations are expected to overlap with the dd-excitations [43].

to now read

Our results on the higher-energy charge excitations call for future RIXS measurements in this regime; however, the high-energy portion of the charge excitations may overlap with the intra-atomic orbital excitations on the Cu atom (the so-called dd-excitations), which are commonly observed at the Cu L-edge [43].

The Reviewer wrote: 5. I suggest replacing the sentence ”(The dynamical spin structure factor $S(Q, \omega)$ and spin susceptibility $\chi_s(Q, \omega)$ satisfy a similar equation)” by the equation itself.

Our response: We took the reviewer’s suggestion and made the change.

The Reviewer wrote: 6. It is very interesting to have access to the codes used for this project. The link provided for both github repository give access to the very general repositories, with multiple branches. It would be necessary to have a tag for the version used for this project. I suggest a tag in the git history, but a fork of the code in separate git repositories, dedicated to the versions used in this manuscript would be even better.

Our response: We took the reviewer’s suggestion and created a fork in a separate repository. The link to the repository is provided in the “code availability” section.

The Reviewer wrote: 7. The equation for local density operator suggests that it is possible to measure local quantity with DCA. Usually DCA is periodized, so the Green function is translationally invariant. I deduce that the local charge operator is then measured on the cluster directly. In the text, I would write, in the sentence after equation 4: “[...] is the local density operator measured on the cluster from the impurity solver.”

Our response: We would first like to clarify that the figures do not plot the local densities but rather the spin-spin and density-density correlation functions. Moreover, all plotted quantities are measured on the cluster directly using the cluster solver, as the Referee correctly deduced. In response to the Reviewer’s comment, we have revised the text following the current Eq. (4) to read:

Here, $\mathbf{r} = (r_x, r_y)a$ and $\mathbf{i} = (i_x, i_y)a$ are lattice vectors and $\hat{S}_i^z = \frac{1}{2}(n_{i,\uparrow} - n_{i,\downarrow})$ and $n_i = \sum_{\sigma} c_{i,\sigma}^{\dagger} c_{i,\sigma}$ are the local spin- z and density operators of the effective cluster problem. The correlation functions $\chi_{c,s}(\mathbf{r}, 0)$ are measured directly by the cluster solver.

The Reviewer wrote: Physics: 1. There is a strong claim that the CDW seen on the e-doped case is not correlated to the SDW, as opposed to the h-doped case. I do not see this clearly from Fig.2. Indeed, Fig.2a and b both have a somewhat misleading colorbar in the sense that most dots on the far left and right have intensity below 3% of the maximum value. This means that, except from the 5 centre most columns, most of the data are noise fluctuations around zero. I do not know how much can be claimed from this and if the dotted lines can be trusted.

Our response: We would like to first emphasize that this claim is not based solely on Fig. 2 but rather on both momentum-space and real-space correlation functions shown in Figs. 1 and 2, respectively. This point is important because detecting the incommensurate peaks in real space is more challenging due to the short correlation lengths at these temperatures. Conversely, the corresponding correlations in momentum space appear as peaks in the relevant correlation functions, which can be easier to extract from the Monte Carlo data. Importantly, our claims are supported by *both* the real and momentum space correlations.

Regarding the signal strength, the small values of the correlation functions toward the edge of the cluster reflect the short correlation length, which produces a rapidly decaying signal in real space. However, the reliability of the data is determined by the signal-to-noise ratio and not the size of the data point relative to the color scale. To demonstrate that the signal is reliable, we have performed additional DCA calculations for the *h*- and *e*-doped cases shown in Fig. 2 of the paper. With this new data, we have carried out a binning analysis to extract the statistical error in the Monte Carlo data. We have now updated Fig. 2 by indicating the sites in the cluster with the sign of the correlation only when the absolute mean of the signal is larger than two standard errors. (For reference, the revised figure is copied here in Fig. 2.) The new plots demonstrate that the modulated charge and spin correlations are robust beyond the 5 center sites. We have also performed a comparable error analysis for the correlation functions in momentum space. The results are shown in the supplementary materials for the lowest accessible temperatures. (For reference, the revised figure is copied here in Fig. 3.) In this case, the incommensurate peak structure of $\chi_c(\mathbf{Q}, 0)$ and $\chi_s(\mathbf{Q}, 0)$ is both clear and stronger than any statistical fluctuations in the data. This analysis makes it clear that our

Figure 2: **Static spin and charge correlations in real-space.** **a** $\chi_c(\mathbf{r}, 0)$ for the e -doped system ($t' = -0.2t$, $\langle n \rangle = 1.2$) at the lowest accessible inverse temperature $\beta t = 10$. **b** $\chi_c(\mathbf{r}, 0)$ for the h -doped system ($t' = -0.2t$, $\langle n \rangle = 0.8$) at the lowest accessible inverse temperature $\beta t = 6$. **c** and **d** show the staggered spin-spin correlations $\chi_{s,\text{stag}}(\mathbf{r}, 0)$ for the h - and e -doped cases, respectively. + and - signs indicate the sign of the correlations whose absolute mean is larger than two standard errors. The dashed lines indicate the approximate nodes in the spin and charge stripe modulations.

data can be trusted and that our conclusions are robust.

The Reviewer wrote: 2. In the text, it is written: “[...] are plotted in Fig. 2, with the data for the h -doped case reproduced from Ref. 34”, which seems to suggest that Fig.2d (and potentially Fig.2c) are taken directly from Ref. 34. I have looked at Ref. 34 and no such data are found, so I deduced that my first reading was wrong and this sentence only meant that “Fig.2d is very similar to Fig.1a of Ref. 34.” I would be much clearer about it. I would even add something like “This similarity act as a benchmark that bridges the results from DQMC method (used in Ref. 10 and 34) with our DCA method.” Although, there is a factor of 5 in the colorscale I cannot explain, this should also be addressed in the text.

Our response: We thank the reviewer for this comment. The old Fig. 2b and Fig. 2d were taken from Fig. 1d and Fig. 2d, respectively, of Ref. 34. In the revised manuscript, the h -doped panels in the updated Fig. 2 are plotted with the regenerated data that includes the error bar. We have, therefore, added a sentence stating that the new results for the h -doped model are indeed very similar to our prior paper. We also note that detailed comparisons of our DCA results against DQMC were performed in Ref. [34].

The color scales in Ref. [34] are the same as those used in the current manuscript. The spin and charge correlations range from ± 0.02 and ± 0.002 in Fig. 2 of the current paper, which is the same as in Figs. 1 (spin) and 2 (charge) of Ref. [34].

The Reviewer wrote: 3. The equations 3 and 4 are really the spatial average of the local cluster spin and charge susceptibility. It is not clear how accurate this averaging is. Indeed, in DCA the self-consistency is done on a periodic and translational invariant Green function. This does not imply that the original local cluster quantities are a good representation of the real physics. This needs to be addressed in the text. Perhaps the best answer to this point is that the results from Ref. 34 are well reproduced, hence this constitutes a good benchmark. Any additional comments or reference would improve the strength of the conclusion.

Our response: As the Reviewer notes, the DCA cluster has periodic boundary conditions and, therefore, all quantities calculated on the cluster, including the spin and charge correlation functions in Eqs. (3) and (4) have translational invariance to within the statistical noise. The averaging in those equations is simply done to reduce the statistical noise in the Monte Carlo data. The same equations were used to calculate the correlation functions in Ref. [34]. The hole-doped case in Ref. [34] with $t' = -0.2$, $\beta = 6$, $\langle n \rangle = 0.8$ is

Figure 3: **Static charge and spin correlations in momentum-space from DCA simulations.** Panels **a** and **b** show the static charge susceptibility $\chi_c(\mathbf{Q}, 0)$ along the $(Q_x, 0)$ direction, for h - and e -doped systems respectively at the their lowest temperatures. Panel **c** and **d** show the corresponding static spin susceptibility $\chi_s(\mathbf{Q}, 0)$ along the $\mathbf{Q} = (Q_x, 0.5)$ direction. These results correspond to the real-space data in Fig. 2 with error bar.

indeed well reproduced within error bars in the revised Fig. 2 (copied here in Fig. 2).

The Reviewer wrote: 4. In the discussion, the authors spend some time arguing why their result differs quantitatively from the experiments. I find this portion superfluous because it is generally known and accepted that CDMFT and DCA provide qualitatively accurate results, but rarely quantitatively accurate. Indeed, the quantitative inaccuracy of the CDW peak compared to the experiment could be caused by the inaccuracy of the experimental doping value because of the annealing method. But many more causes are possible, principally the general numerical inaccuracy of the Hubbard model solved with DMFT extensions and their approximations.

Our response: We agree that cluster methods like CDMFT or DCA may only provide qualitatively accurate results and that systematic errors may also contribute to the disagreement between the observed and predicted values of \mathbf{Q}_{cdw} . In fact, we explicitly acknowledged that the results obtained from DCA have quantitative differences to other nonperturbative methods, stating

Finally, the periodicity of the charge modulations may be affected by the DCA mean-field [34].

This sentence is citing systematic comparisons we performed between the results obtained from DCA and DQMC simulations of the h -doped model.

We respectfully disagree that the discussion of the uncertainties in the doping levels quoted by the RIXS experiments is superfluous. Such uncertainties can certainly contribute to disagreements between the theory and experiment, so it is important to be aware of them. Moreover, the uncertainty in the doping level affects our ability to confidently determine if the observed charge stripes are filled. This point was raised by the first Reviewer, so it highlights the need to make a general reader aware of this issue.

REVIEWERS' COMMENTS

Reviewer #1 (Remarks to the Author):

I appreciate the authors' efforts to answer most of my questions. It is now clearer to me that the results of the electron-doped cases at finite temperatures are novel. While I agree that a precise one-to-one comparison between the Hubbard model with t' and the t-J model may not be possible due to the relatively modest value of U used in the current study, it appears that $U=6t$ is still relatively close to the bandwidth of the system, meaning that the strong correlation effect cannot be neglected in the Hubbard model. Therefore, a qualitative comparison between the Hubbard model and the t-J model would be reasonable. There are several recent studies on the t-J model in the electron-doped cases with positive t' and a whole electron density smaller than half-filling. For example, Ref. [12] reports on such cases in PNAS 118, e2109978118 (2021). Although a quantitative comparison may not be feasible, this study could be qualitatively compared to the current electron-doped Hubbard study after a particle-hole transformation. However, I did not notice the report of a checkboard component of charge density wave order in this case, including the 20% doping case. It is possible that this discrepancy could be due to differences in the system sizes and effective U between Ref. [12] and the current study. In my opinion, a qualitative comparison is not entirely unreasonable and would be worth considering, as it could further improve the reliability of the current study.

With regards to the relationship between the present study and the RIXS experiment, the authors have raised questions about the reference I used in my previous report. The reference in question is Ref. [28], which is depicted in Fig. 3c. Having read the authors' response letter, I took a closer look at the RIXS data points and carefully estimated its slope. As a result, I now believe that the data is more consistent with the filled stripe as opposed to the half-filled charge stripe. I am convinced by the numerical results and their connection to the RIXS experiment.

I am now pleased to recommend the publication of this study. While I still believe that a (qualitative) comparison between the electron-doped t-J model and the Hubbard model is worth considering, as it could further enhance the reliability of the study, it may not be considered as critical, as they are distinct at a quantitative level, particularly when U is smaller than the bandwidth.

Reviewer #2 (Remarks to the Author):

I have read all the changes done to the manuscript. I consider that the reply satisfies each of my concerns in my previous report. The quality of the manuscript has been much improved.

I must insist however that confirming that the Hubbard model is good enough for the e-doped cuprate is not really new. But I agree that this is one new and independent confirmation.